# DOGS: Distributed-Oriented Gaussian Splatting for Large-Scale 3D Reconstruction Via Gaussian Consensus

**Yu Chen**
National University of Singapore
yuchen01@u.nus.edu

**Gim Hee Lee**
National University of Singapore
gimhee.lee@nus.edu.sg

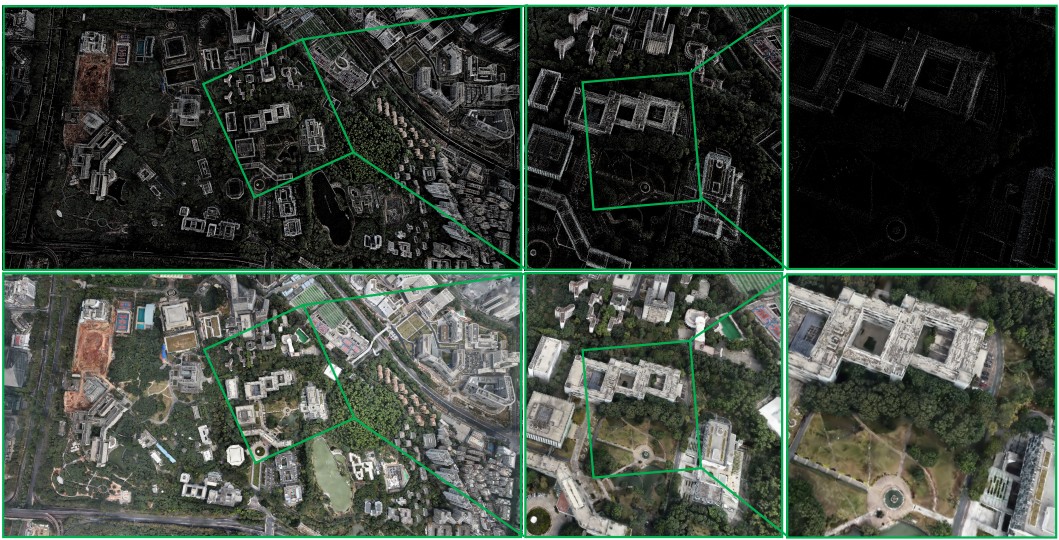

Figure 1: DOGS accelerates 3D GS training on large-scale UrbanScene3D dataset [29] by $6+$ times with better rendering quality. Top: 3D Gaussian primitives (8.27M); Bottom: rendered images.

## Abstract

The recent advances in 3D Gaussian Splatting (3DGS) show promising results on the novel view synthesis (NVS) task. With its superior rendering performance and high-fidelity rendering quality, 3DGS is excelling at its previous NeRF counterparts. The most recent 3DGS method focuses either on improving the instability of rendering efficiency or reducing the model size. On the other hand, the training efficiency of 3DGS on large-scale scenes has not gained much attention. In this work, we propose DOGS, a method that trains 3DGS distributedly. Our method first decomposes a scene into $K$ blocks and then introduces the Alternating Direction Method of Multipliers (ADMM) into the training procedure of 3DGS. During training, our DOGS maintains one global 3DGS model on the master node and $K$ local 3DGS models on the slave nodes. The $K$ local 3DGS models are dropped after training and we only query the global 3DGS model during inference. The training time is reduced by scene decomposition, and the training convergence and stability are guaranteed through the consensus on the shared 3D Gaussians. Our method accelerates the training of 3DGS by $6+$ times when evaluated on large-scale scenes while concurrently achieving state-of-the-art rendering quality. Our code is publicly available at https://github.com/AIBluefisher/DOGS.

38th Conference on Neural Information Processing Systems (NeurIPS 2024).

# 1 Introduction

Neural 3D scene reconstruction has taken a giant step beyond the limitations of traditional photogrammetry tools. Neural radiance fields (NeRFs) [36, 17, 47], which encode scenes implicitly in MLP or explicitly in voxels, exhibit superior resilience to non-Lambertian effects, appearance changes, dynamic scenes, *etc*. However, most NeRF-based methods are inefficient in rendering scenes due to the need to query massive points for volume rendering to infer scene geometry and color. Recently, 3D Gaussian Splatting (3DGS) [21] has shown promising results on real-time applications and inspires many follow-up works. 3DGS encodes scenes into a set of 3D anisotropic Gaussians, where each 3D Gaussian is represented by a covariance matrix, a center position, opacity, and the latent features to encode color information. Pixel colors by projecting 3D Gaussians into 2D image space can be efficiently computed via rasterization, which is highly optimized on modern graphics processors. However, 3DGS often requires larger memory during training compared to NeRF methods. This is because 3DGS needs millions of 3D Gaussians to represent a scene to recover high-fidelity scene details. Consequently, the memory footprint increases drastically for training 3DGS on larger scenes, *e.g.* city-scale scenes. Moreover, the training of a huge number of 3D Gaussians on larger scenes leads to longer training time. Unfortunately, in comparison to NeRF where the rays can be conveniently distributed into different compute nodes, dispatching 3D Gaussians into different compute nodes is much more difficult due to the highly customized rasterization procedure of 3DGS. In summary, the ***two challenges*** for reconstructing large-scale scenes with 3DGS are: 1) High GPU memory to hold the large 3D model during training; 2) Long training time due to the large areas of scenes.

Previous large-scale NeRF methods [49, 51, 35] solve the above-mentioned two issues by embracing a divide-and-conquer approach, where scenes are split into individual blocks with smaller models fitted into each block. However, these methods require querying multiple sub-models during inference, which slows down the rendering efficiency. This leads us to the following question:

"*Can we apply a similar methodology to 3DGS during training while querying only a global consistent model during inference?*"

In this work, we propose DOGS to answer the aforementioned question. Following previous large-scale NeRF methods, our DOGS splits scene structures into blocks for distributed training. Inspired by previously distributed bundle adjustment methods [15, 62], we apply the Alternating Direction Method of Multipliers (ADMM) [5] to ensure the consistency of the shared 3D Gaussians between different blocks. Specifically, we first estimate a tight bounding box for each scene. Subsequently, we split training views and point clouds into blocks. To guarantee each block has a similar size, we split scenes into two blocks each time along the longer axis of the bounding box. Scenes are split recursively in the same way until we obtain the desired number of blocks. Finally, we re-estimate a bounding box for each block and expand the bounding box to construct shared 3D Gaussians between blocks. During training, we maintain a global 3D Gaussian model on the master node and dispatch local 3D Gaussians into other slave nodes. We further guarantee the consistency of the shared 3D Gaussians and the convergence of the training through 3D Gaussian consensus. Specifically, the local 3D Gaussians are collected onto the master node and averaged to update the current global 3D Gaussian model at the end of each training iteration, and then the updated global 3D Gaussian model is shared to all slave nodes to regularize the training of local 3D Gaussians. In this way, our method guarantees that the local 3D Gaussians converge to the global 3D Gaussian model during training. By training 3DGS in a distributed way, our DOGS can reduce the training time by 6+ times compared to the original 3DGS. Furthermore, our DOGS guarantees training convergence and therefore achieves better rendering quality than its counterparts with the 3D Gaussians consensus in the distributed training. After training, we can drop all local 3D Gaussians and maintain only the global 3D Gaussians. During inference, we only need to query the global model while maintaining the rendering performance of 3DGS.

Our contributions are summarized as follows:

- We propose a recursive approach to split scenes into blocks with balanced sizes.

- We introduce a distributed approach to train 3D Gaussians for large-scale scenes. The training time is reduced by $6+$ times compared to the original 3DGS.

- We conduct exhaustive experiments on standard large-scale datasets to validate the effectiveness of our method.

## 2  Related Work

**Neural Radiance Fields.**   Neural radiance fields [36] enable rendering from novel viewpoints with encoded frequency features [50]. To improve training and rendering efficiency, follow-up works [27, 59, 17] either encodes scenes into sparse voxels, or a multi-resolution hash table [37] where the hash collision is implicitly handled during optimization. TensoRF [6] uses CP-decomposition or VM-decomposition to encode scenes into three orthogonal axes and planes. While the previously mentioned work focuses on per-scene reconstruction, other methods also focus on the generalizability of NeRF  [54, 7, 19, 46, 28], bundle-adjusting camera poses and NeRF [25, 34, 8], and leveraging sparse or dense depth to supervise the training of NeRF [56, 14, 42], *etc*. To address the aliasing issue in vanilla NeRF, Mip-NeRF [1] proposed to use Gaussian to approximate the cone sampling, the integrated positional encodings are therefore scale-aware and can be used to address the aliasing issue of NeRF. Mip-NeRF360 [2] further uses space contraction to model unbounded scenes. Zip-NeRF [3] adopted a hexagonal sampling strategy to handle the aliasing issue for Instant-NGP [37]. NeRF2NeRF [18] and DReg-NeRF [9] assumes images are only available during training in each block, and they propose methods to register NeRF blocks together.

**Gaussian Splatting.**   Gaussian Splatting [21] initializes 3D Gaussians from a sparse point cloud. The 3D Gaussians are used as explicit scene representation and dynamically merged and split during training. Real-time novel view synthesis is enabled by the fast splatting operation through rasterization. Scaffold-GS [31] initializes a sparse voxel grid from the initial point cloud and encodes the features of 3D Gaussians into corresponding feature vectors. The introduction of the sparse voxel reduced the Gaussian densities by avoiding unnecessary densification on the scene surface. Octree-GS [41] introduces the level-of-details to dynamically select the appropriate level from the set of multi-resolution anchor points, which ensures consistent rendering performance with adaptive LOD adjustment and maintains high-fidelity rendering results. To reduce the model size, methods [16, 38] also remove redundant 3D Gaussians through exhaustive quantization. Other methods also focus on alleviating the aliasing issue of 3D Gaussians [60, 58], or leveraging the efficient rasterizer of point rendering for real-time indoor reconstruction [20, 33].

**Large-Scale 3D Reconstruction.**   The classical photogrammetry methods utilized Structure-from-Motion (SfM) [43] and keypoint extraction with SIFT [30] to reconstruct sparse scene structures. One of such foundational software is Phototourism [45]. To handle city-scale scenes, the 'divide-and-conquer' strategy is widely adopted for the extensibility and scalability of the 3D reconstruction system. SfM methods [4, 63, 10, 12, 11] splitting scenes based on the view graph, where images with strong connections are divided into the same block. By estimating similarity transformations and merging tracks, all local SfM points and camera poses are fused into a global reference system. Existing NeRF methods also follow a similar strategy for reconstructing city-scale scenes. When camera poses are known, the scene can be split into grid cells. Block-NeRF [49] focus on the day-night or even cross-season street views. It utilizes the appearance encoding in NeRF-W [32] to fix the appearance inconsistency issue between different blocks, while Mega-NeRF [51] aims at encouraging the sparsity of the network under aerial scenes. Urban-NeRF [40] leverages lidar points to supervise the depth of NeRF in outdoor scenes. SUDS [52] further extended Mega-NeRF into dynamic scenes. Different from previous large-scale NeRF methods, Switch-NeRF [35] uses a switch transformer that learns to assign rays to different blocks during training. Grid-NeRF [57] designed a two-branch network architecture, where the NeRF branch can encourage the feature plane [6] branch recover more scene details under large-scale scenes. However, the two-branch training scheme is trivial and needs a long time to train. Concurrent works to our method are VastGaussian [26] and Hierarchy-GS [22], which also utilize 3D Gaussians for large-scale scene reconstruction. VastGaussian and Hierarchy-GS split scenes into independent chunks and train independent chunks simultaneously. However, VastGaussian relies on exhaustive searching of the training views and initial points to guarantee the convergence of training, and each block is trained without data sharing. Hierarchy-GS consolidates independent chunks into intermediate nodes for further rendering. However, the hierarchical approach needs to preserve redundant models and it is specially designed for street view scenes. Our method, on the other hand, focuses on the distributed training of 3DGS and built upon the consensus of shared 3D Gaussians between different blocks has a guaranteed convergence rate that achieves better performance.

## 3 Our Method

### 3.1 Preliminary

**Gaussian Splatting.** 3D Gaussian Splatting represents a scene with a set of anisotropic 3D Gaussians $\mathcal{G} = \{\mathbf{G}_i \mid i \in N\}$. Each 3D Gaussian primitive $\mathbf{G}_i$ has a center $\mathbf{u} \in \mathbb{R}^3$ and a covariance $\mathbf{\Sigma} \in \mathbb{R}^{3 \times 3}$ and can be described by:

$$\mathbf{G}_i(\mathbf{p}) = \exp\{-\frac{1}{2}(\mathbf{p} - \mathbf{u}_i)^\top \mathbf{\Sigma}_i^{-1}(\mathbf{p} - \mathbf{u}_i)\}. \tag{1}$$

During training, the covariance is decomposed into a rotation matrix $\mathbf{R} \in \mathbb{R}^{3 \times 3}$ and a diagonal scaling matrix $\mathbf{S} \in \mathbb{R}^{3 \times 3}$, *i.e.* $\mathbf{\Sigma}_i = \mathbf{R}\mathbf{S}\mathbf{S}^\top\mathbf{R}^\top$ to ensure the covariance matrix is positive semi-definite. To render the color for a pixel $\mathbf{p}$, the 3D Gaussians are projected into the image space for alpha blending:

$$\mathbf{C} = \sum_i \mathbf{c}_i \alpha_i \prod_{j=1}^{i-1}(1 - \alpha_j), \tag{2}$$

where $\alpha_i$ is the rendering opacity and is computed by $\alpha = o \cdot \mathbf{X}^{\text{proj}}(\mathbf{p})$.

When training 3D Gaussian Splatting, we minimize the loss function below:

$$\mathcal{L}(\mathbf{x}) = \mathcal{L}_{\text{rgb}} + \lambda \mathcal{L}_{\text{ssim}}, \tag{3}$$

where $\mathbf{x}_i = \{\mathbf{u}_i, \mathbf{q}_i, \mathbf{s}_i, \mathbf{f}_i, o_i\}$, $\mathbf{q}$ are quaternions corresponds to the rotation matrix $\mathbf{R}$, $\mathbf{s}$ are vectors corresponds to the three diagonal elements of $\mathbf{S}$, and $\mathbf{f}$ are coefficients of the spherical harmonics.

### 3.2 Distributed 3D Gaussian Consensus

The 'divide-and-conquer' method is a common paradigm for large-scale 3D reconstruction, which we also adopt in our framework. Different from previous methods such as Block-NeRF [49] and VastGaussian [26] which are pipeline parallelized, our method is optimization parallelized with guaranteed convergence. The pipeline of our algorithm is shown in Fig. 2. Firstly, we split a scene (training views and point clouds) into $K$ intersected blocks. Secondly, we assign training views and points into different blocks. By introducing the ADMM into the training process, we also maintain a globally consistent 3DGS model on a master node. Thirdly, during training, we collect and average the local 3D Gaussians to update the global 3DGS model in each consensus step. The global 3D Gaussians are also shared with each block before we distributedly train the local 3D Gaussians in each block. Finally, we drop all local 3D Gaussians while only using the global 3D Gaussians to render novel views.

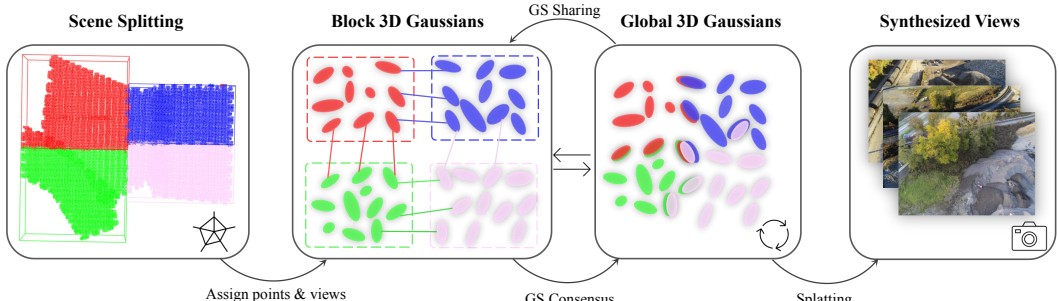

Figure 2: **The pipeline of our distributed 3D Gaussian Splatting method**. 1) We first split the scene into $K$ blocks with similar sizes. Each block is extended to a larger size to construct overlapping parts. 2) Subsequently, we assign views and points into different blocks. The shared local 3D Gaussians (connected by solid lines in the figure) are a copy of the global 3D Gaussians. 3) The local 3D Gaussians are then collected and averaged to the global 3D Gaussians in each consensus step, and the global 3D Gaussians are shared with each block before training all blocks. 4) Finally, we use the final global 3D Gaussians to synthesize novel views.

In this section, we first introduce the ADMM algorithm. Subsequently, we derive the distributed 3DGS training algorithm. We also present a scene splitting algorithm, which recursively and evenly splits the scene into two blocks each time.

**ADMM.** A general form for consensus ADMM is given by:

$$\text{minimize} \sum_{i=1}^{N} f_i(\mathbf{x}_i), \quad \text{s.t. } \mathbf{x}_i - \mathbf{z} = 0, \ i \in [1, N]. \tag{4}$$

By definition, the constraints are applied such that all the local variables $\mathbf{x}_i$ agree with the global variable $\mathbf{z}$. By applying the augmented Lagrangian method, we have:

$$\mathcal{L}_\rho(\mathbf{x}, \mathbf{z}, \mathbf{y}) = \sum_{i=1}^{N} \left( f_i(\mathbf{x}_i) + \mathbf{y}_i^\top (\mathbf{x}_i - \mathbf{z}) + \frac{\rho}{2} \|\mathbf{x}_i - \mathbf{z}\|_2^2 \right), \tag{5}$$

where $\mathbf{y}_i$ is the dual variable, $\rho$ is the penalty parameter. During optimization, ADMM alternatively updates the local variables $\mathbf{x}_i$, global variable $\mathbf{z}$ and the dual variables $\mathbf{y}_i$ at the $t + 1$ iteration by:

$$\mathbf{x}_i^{t+1} := \arg\min \left( f_i(\mathbf{x}_i) + \mathbf{y}_i^{t\top} (\mathbf{x}_i - \mathbf{z}^t) + \frac{\rho}{2} \|\mathbf{x}_i - \mathbf{z}^t\|_2^2 \right), \tag{6a}$$

$$\mathbf{z}^{t+1} := \frac{1}{N} \sum_{i=1}^{N} \left( \mathbf{x}_i^{t+1} + \frac{1}{\rho} \mathbf{y}_i^t \right), \tag{6b}$$

$$\mathbf{y}_i^{t+1} := \mathbf{y}_i^t + \rho(\mathbf{x}_i^{t+1} - \mathbf{z}^{t+1}). \tag{6c}$$

**Distributed Training.** We apply the ADMM method to distributedly train a large-scale 3D Gaussian Splatting model. In our problem, $f_i(\cdot)$ in Eq. (4) corresponds to the loss function $\mathcal{L}(\cdot)$ in Eq. (3). To simplify the implementation for Eq. (6a), we adopt a scaled form of ADMM by defining $\mathbf{u}_i = \frac{1}{\rho} \mathbf{y}_i$. We can then rewrite Eq. (5) into (see supplementary for the derivation):

$$\mathcal{L}_\rho(\mathbf{x}, \mathbf{z}, \mathbf{u}) = \sum_{i=1}^{N} \left( f_i(\mathbf{x}_i) + \frac{\rho}{2} \|\mathbf{x}_i - \mathbf{z} + \mathbf{u}_i\|_2^2 - \frac{\rho}{2} \|\mathbf{u}_i\|_2^2 \right). \tag{7}$$

Compared to Eq. (5), Eq. (7) can be made easier to implement by expressing all terms in the squared difference errors. Suppose the variables are decomposed into $K$ blocks, we then denote the $i$th 3D Gaussians in the $k$th block as $\mathbf{x}_i^k$. Accordingly, we revise the ADMM updating rule by:

$$(\mathbf{x}_i^k)^{t+1} := \arg\min \left( f(\mathbf{x}_i^k) + \frac{\rho}{2} \|(\mathbf{x}_i^k)^t - \mathbf{z}^t + (\mathbf{u}_i^k)^t\|_2^2 \right), \tag{8a}$$

$$\mathbf{z}^{t+1} := \frac{1}{K} \sum_{k=1}^{K} (\mathbf{x}_i^k)^{t+1}, \tag{8b}$$

$$(\mathbf{u}_i^k)^{t+1} := (\mathbf{u}_i^k)^t + (\mathbf{x}_i^k)^{t+1} - \mathbf{z}^{t+1}. \tag{8c}$$

Eq. (8a) is the original loss function in Gaussian Splatting with an additional regularizer term $\frac{\rho}{2} \|(\mathbf{x}_i^k)^t - \mathbf{z}^t + (\mathbf{u}_i^k)^t\|_2^2$. Note that Eq. (8b) should be $\mathbf{z}^{t+1} := \frac{1}{K} \sum_{k=1}^{K} \left( (\mathbf{x}_i^k)^{t+1} + (\mathbf{u}_i^k)^t \right)$. However, the dual variables have an average value of zero after the first iteration. Consequently, Eq. (8b) can be simplified as the global 3D Gaussians are formed by the average of the local 3D Gaussians from all blocks. Moreover, Eq. (8b) is called a 'consensus' step and it requires collecting the local 3D Gaussians from all blocks. After updating the global model $\mathbf{z}$, we update the dual variables $\mathbf{u}_i$ in Eq. (8c) and share the global 3D Gaussians $\mathbf{z}$ to each block for optimizing the local 3D Gaussians in Eq. (8a). Note that each 3D Gaussian $\mathbf{x}_i$ has different properties $\{\mathbf{u}_i, \mathbf{q}_i, \mathbf{s}_i, \mathbf{f}_i, o_i\}$. As a result, the penalty terms and dual variables should be represented separately according to these properties. The detailed form of Eq. (8) is given in the supplementary.

**Scene Splitting.** We decompose the scene into $K$ blocks before applying the updating rule in Eq. (8). Unlike VastGaussian [26], which focuses mostly on splitting large-scale scenes and needs exhaustive search on the training views and point clouds to ensure the consistency of 3D Gaussians in different blocks, our method relies on the consensus step to ensure the consistency of the 3DGS. However, scene splitting is still important to the convergence of our method. We propose two constraints for the scene-splitting method to best balance the training efficiency and rendering quality:

1. Individual blocks should have a similar size.

2. Adjacent blocks should have enough overlaps to boost the convergence of training.

The first constraint is proposed to ensure that: 1) Each block can be fed into GPUs with the same capacity. This is important since a larger block can cause an out-of-memory of the GPU during training due to the imbalanced splitting results. 2) Each block has a similar training time at each iteration. After every $t$ iteration, we collect all local 3D Gaussians from each block. Intuitively, larger blocks require a longer time to train. Consequently, the master node and all other slave nodes have to wait for the nodes with larger blocks to finish, which increases the training time unnecessarily.

The second constraint is used to boost the convergence of ADMM. From Eq. (8b), the local 3D Gaussians would converge to the global 3D Gaussians by averaging the shared local 3D Gaussians during training. Sufficient shared 3D Gaussians encourage reconstruction consistency between different blocks. Too many shared local 3D Gaussians can bring more communication overhead, which inevitably slows down the training while a lack of shared 3D Gaussians leads to divergence of the algorithm. Although there is no theoretical analysis to show the optimal value of overlapping parts, we empirically use a constant value which we will introduce later in our experiments.

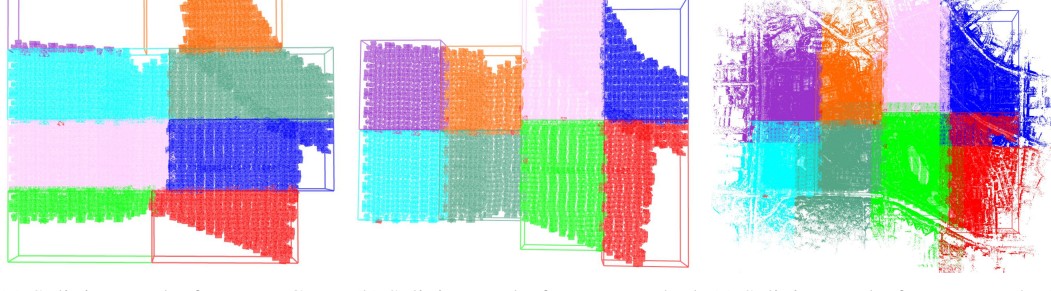

(a) Splitting results from VastGaussian (camera trajectories)  (b) Splitting results from our method (camera trajectories)  (c) Splitting results from our method (camera trajectories and 3D points)

Figure 3: **Scene splitting results of our method v.s. VastGaussian [26]**. (a) VastGaussian can result in imbalanced blocks. (b) Our recursive bipartite strategy solves the imbalanced splitting issue. (c) Points and views with the same grid coordinate are assigned to the same block.

We assume one of the axes of the scene is aligned to physical ground, which can usually be done under the Manhattan world assumption. VastGaussian [26] adopts a grid-splitting method that first splits the scene into $m$ cells along the x-axis, and then splits each of the $m$ cells into $n$ sub-cells along the y-axis. As we show in Fig. 3a, this strategy can result in imbalanced blocks. Our splitting method is inherited from VastGaussian while adopting a recursive spitting method to resolve the imbalanced issue. Specifically, we first estimate a tight bounding box for the initial scene. We then split the scene into two parts along the longer axis of the scene. Splitting the scene along the longer axis can prevent the blocks from becoming too shallow on one axis. We re-estimate a tighter bounding box for each of the two cells and split them into smaller blocks. This step is repeated until the number of blocks reaches our requirement. We present the result of this recursive method in Fig. 3b. Compared to Fig. 3a, we produce more balanced blocks. To construct overlapping areas, we expand the bounding box of each block by a scale factor $s$, any points/views that are covered by the same bounding box are grouped into the same block. The training views and point clouds are split in the same way as is shown in Fig. 3c.

### 3.3 Improving Convergence Rate

ADMM is known to be sensitive to the initialization of penalty parameters. Since improper initial penalty parameters can slow down the training, we introduce the *adaptive penalty parameters* and *over-relaxation* to improve the convergence rate.

**Primal Residual and Dual Residual.** We define the primal residual $\mathbf{r}^t$ and the dual residual $\mathbf{s}^t$ as

$$\mathbf{r}^t = \mathbf{x}_i^t - \mathbf{z}^t, \quad \mathbf{s}^t = \rho(\mathbf{z}^t - \mathbf{z}^{t-1}). \tag{9}$$

In ADMM, the primal residual and dual residual are used as the stopping criteria which terminate the optimization. In our method, we use a hard threshold of training iteration to terminate the algorithm. The primal residual and dual residual are used to adaptively adjust the penalty parameters.

**Adaptive Penalty Parameters.** We adopt a similar scheme from [5] to adaptively adjust the penalty parameters:

$$\rho^{t+1} = \begin{cases} \tau^{\text{inc}} \rho^t, & \|\mathbf{r}^t\|_2 > \mu \|\mathbf{s}^t\|_2, \\ \dfrac{\rho^t}{\tau^{\text{dec}}}, & \|\mathbf{s}^t\|_2 > \mu \|\mathbf{r}^k\|_2, \\ \rho^t, & \|\mathbf{s}^t\|_2 = \mu \|\mathbf{r}^k\|_2, \end{cases} \tag{10}$$

where $\mu > 1$, $\tau^{\text{inc}}$, $\tau^{\text{dec}}$ are hyper-parameters. The existing convergence proof of the ADMM algorithm is based on the fixed penalty parameters [5]. To guarantee the convergence of our algorithm, we stop adjusting the penalty parameters after 2000 iterations in all of our experiments.

**Over Relaxation.** Similar to [5], we replace $\mathbf{x}^{t+1}$ with $\alpha^t \mathbf{x}^{t+1} - (1 - \alpha^t)(-\mathbf{z}^t)$ in Eq. (8b) and Eq. (8c), where $\alpha^t \in (0, 2)$ is the relaxation parameter and experiments show that the over-relaxation with $\alpha^t \in [1.5, 1.8]$ can improve convergence.

## 4   Experiments

**Datasets.** We evaluate our method on the two large-scale urban datasets, the Mill19 [51][1] dataset, the UrbanScene3D [29][2] dataset, and the MatrixCity [24] dataset. Both datasets are captured by drones and each scene contains thousands of high-resolution images. During training and evaluation, we adopt the original image splitting in Mega-NeRF [51], and downsample the images by 6+ times from the original resolution.

**Implementation Details.** For Mill19 and UrbanScene3D, we use the camera poses provided by the official site of Mega-NeRF [51]. The y-axis of the scene is aligned to the horizontal plane by COLMAP [43] under the Manhattan world assumption. We use the CPU version of SIFT (SIFT-CPU) in COLMAP [43] to extract keypoints and match each image to its nearest 100 images using vocabulary trees. With known camera poses and keypoint matches, we further triangulate 3D points and bundle adjust them. The SIFT-CPU can extract more points than the SIFT-GPU, which can benefit the initialization of 3D Gaussians. For the original 3D Gaussian Splatting (denoted as '3DGS'), we train it in 500,000 iterations and densify it for every 200 iterations until it reaches 30,000 steps. We train both VastGaussian and our method in 80,000 iterations. The densification intervals and termination steps are the same as the 3DGS. We reimplement VastGaussian [26] since its code was not released during this work. Note that we did not implement the decoupled appearance embedding in VastGaussian, which can be used to remove floaters caused by inconsistent image appearance. We argue that we still provided a fair comparison since this module can be applied to all 3DGS-based methods. For our method, consensus and sharing are enabled every 100 iterations. We leverage the remote procedure call (RPC) framework of PyTorch [39] to implement our distributed algorithm and transmit data across different compute nodes.

**Results.** We employ PSNR, SSIM [55] and LPIPS [61] as metrics for novel view synthesis. We compare our method against the state-of-the-art large-scale NeRF-based methods: Mega-NeRF [51], Switch-NeRF [35][3], and 3DGS-based methods: 3DGS [21][4], Fed3DGS [48], VastGaussian [26], Hierarchy-GS [22]. For Mega-NeRF and Switch-NeRF, we use the officially provided checkpoints on 8 blocks for evaluation. The results of Hierarchy-GS are cited from the original paper since its code has not been released during this work.

We present the quantitative visual quality results in Table 1. The training time and rendering efficiency are also provided in Table 2. As shown in the tables, methods based on 3D Gaussians achieved better results than NeRF-based methods. Although NeRF-based methods are comparable to 3DGS methods in PSNR, the rendered images lack details in such large-scale scenes. This is also validated from

---

[1] https://github.com/cmusatyalab/mega-nerf
[2] https://vcc.tech/UrbanScene3D
[3] https://github.com/MiZhenxing/Switch-NeRF
[4] https://github.com/graphdeco-inria/gaussian-splatting

Fig. 4 and Fig. 5. Moreover, NeRF-based methods are much slower than 3DGS-based methods and take longer time to train – even if they are trained distributedly. Notably, our method achieves the best results in almost all the scenes. The original 3DGS has results comparable to ours. However, it takes $6 \sim 8\times$ more training time than our method. We also emphasize that we build a strong baseline of 3DGS for fair comparison: the densification interval is 200 iterations, which is 8 times more frequent than the 3DGS baseline in VastGaussian; the training iteration is 500K (in comparison, the training iteration of the 3DGS baseline is 450K in VastGaussian). VastGaussian trains 3DGS faster than our method. This is because our method requires additional time for data transmission. However, our method achieves better rendering quality than VastGaussian. Moreover, the data transmission time does not become the bottleneck of our method due to our balanced scene splitting method. Particularly, each block has a similar training time and the master node does not need to wait for a long time for the fat nodes to finish its job.

To further show the applicability of our method to larger-scale scenes, we evaluated our method on the $2.7\text{km}^2$ Small City scene in the MatrixCity [24] dataset, which contains $5,620$ training views and $741$ validation views. We early terminated the training of the original 3DGS since it did not finish the training within two days. VastGaussian failed on this dataset since two blocks produce no 3D Gaussian primitives due to the imbalanced splitting. From Table 3, our method achieved the best results in rendering quality. The visual qualitative results are shown in Fig. 6.

Table 1: **Quantitative results of novel view synthesis on Mill19 [51] dataset and Urban-Scene3D [29] dataset**. ↑: higher is better, ↓: lower is better. The red, orange and yellow colors respectively denote the best, the second best, and the third best results. † denotes without applying the decoupled appearance encoding.

| Scenes | Building | | | Rubble | | | Campus | | | Residence | | | Sci-Art | | |
|---|---|---|---|---|---|---|---|---|---|---|---|---|---|---|---|
| | PSNR ↑ | SSIM ↑ | LPIPS ↓ | PSNR ↑ | SSIM ↑ | LPIPS ↓ | PSNR ↑ | SSIM ↑ | LPIPS ↓ | PSNR ↑ | SSIM ↑ | LPIPS ↓ | PSNR ↑ | SSIM ↑ | LPIPS ↓ |
| Mega-NeRF [51] | 20.92 | 0.547 | 0.454 | 24.06 | 0.553 | 0.508 | 23.42 | 0.537 | 0.636 | 22.08 | 0.628 | 0.401 | 25.60 | 0.770 | 0.312 |
| Switch-NeRF [35] | 21.54 | 0.579 | 0.397 | 24.31 | 0.562 | 0.478 | 23.62 | 0.541 | 0.616 | 22.57 | 0.654 | 0.352 | 26.51 | 0.795 | 0.271 |
| 3DGS [21] | 22.53 | 0.738 | 0.214 | 25.51 | 0.725 | 0.316 | 23.67 | 0.688 | 0.347 | 22.36 | 0.745 | 0.247 | 24.13 | 0.791 | 0.262 |
| Fed3DGS [48] | 18.66 | 0.602 | 0.362 | 20.62 | 0.588 | 0.437 | 21.64 | 0.635 | 0.436 | 20.00 | 0.665 | 0.344 | 21.03 | 0.730 | 0.335 |
| VastGaussian† [26] | 21.80 | 0.728 | 0.225 | 25.20 | 0.742 | 0.264 | 23.82 | 0.695 | 0.329 | 21.01 | 0.699 | 0.261 | 22.64 | 0.761 | 0.261 |
| Hierarchy-GS [22] | 21.52 | 0.723 | 0.297 | 24.64 | 0.755 | 0.284 | – | – | – | – | – | – | – | – | – |
| DOGS | 22.73 | 0.759 | 0.204 | 25.78 | 0.765 | 0.257 | 24.01 | 0.681 | 0.377 | 21.94 | 0.740 | 0.244 | 24.42 | 0.804 | 0.219 |

Table 2: **Quantitative results of novel view synthesis on Mill19 dataset and UrbanScene3D dataset**. We present the training time (hh:mm), the number of final points ($10^6$), the allocated memory (GB), and the framerate (FPS) during evaluation. † denotes without applying the decoupled appearance encoding.

| Scenes | Building | | | | Rubble | | | | Campus | | | | Residence | | | | Sci-Art | | | |
|---|---|---|---|---|---|---|---|---|---|---|---|---|---|---|---|---|---|---|---|---|
| | Train ↓ | Points | Mem | FPS ↑ | Train ↓ | Points | Mem | FPS ↑ | Train ↓ | Points | Mem | FPS ↑ | Train ↓ | Points | Mem | FPS ↑ | Train ↓ | Points | Mem | FPS ↑ |
| Mega-NeRF [51] | 19:49 | – | 5.84 | 0.009 | 30:48 | – | 5.88 | 0.009 | 29:03 | – | 5.86 | 0.008 | 27:20 | – | 5.99 | 0.006 | 27:39 | – | 5.97 | 0.006 |
| Switch-NeRF [35] | 24:46 | – | 5.84 | 0.009 | 38:30 | – | 5.87 | 0.009 | 36:19 | – | 5.85 | 0.007 | 35:11 | – | 5.94 | 0.007 | 34:34 | – | 5.92 | 0.008 |
| 3DGS [21] | 21:37 | 7.99 | 4.62 | 90.09 | 18:40 | 3.85 | 2.18 | 166.67 | 23:03 | 13.6 | 7.69 | 59.52 | 23:13 | 5.35 | 3.23 | 142.86 | 21:33 | 2.31 | 1.61 | 240.96 |
| VastGaussian† [26] | 03:26 | 5.60 | 3.07 | 121.35 | 02:30 | 4.71 | 2.74 | 163.93 | 03:33 | 17.6 | 9.61 | 47.84 | 03:12 | 6.26 | 3.67 | 118.48 | 02:33 | 4.21 | 3.54 | 120.33 |
| DOGS | 03:51 | 6.89 | 3.39 | 122.33 | 02:25 | 4.74 | 2.54 | 147.06 | 04:15 | 8.27 | 4.29 | 99.85 | 04:33 | 7.64 | 6.11 | 82.34 | 04:23 | 5.67 | 3.53 | 107.87 |

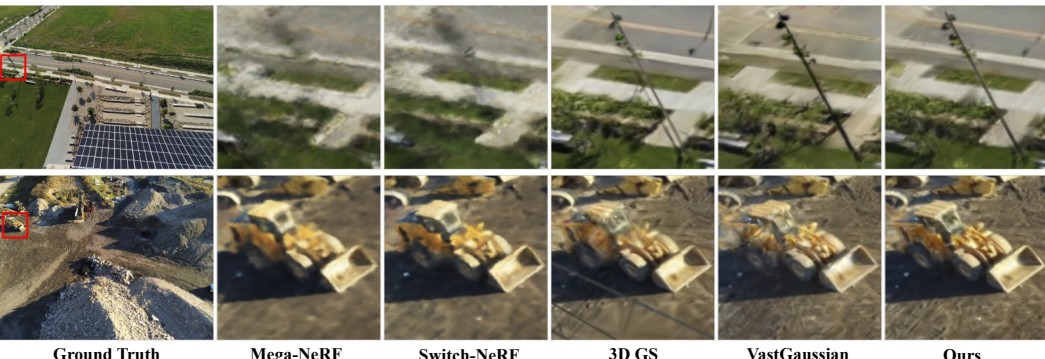

| Ground Truth | Mega-NeRF | Switch-NeRF | 3D GS | VastGaussian | Ours |

Figure 4: **Qualitative comparisons of our method and others on the Mill19 dataset**. The first row and second row are respectively the results of scene 'building' and 'rubble'.

**Ablation Study.** We ablate the effectiveness of our method and present the results in Table 4. Our method without applying the 3D Gaussian consensus is denoted as *w.o. CS*, our method without adopting the self-adaptation of penalty parameters is denoted as *w.o. SD*, our method without adopting

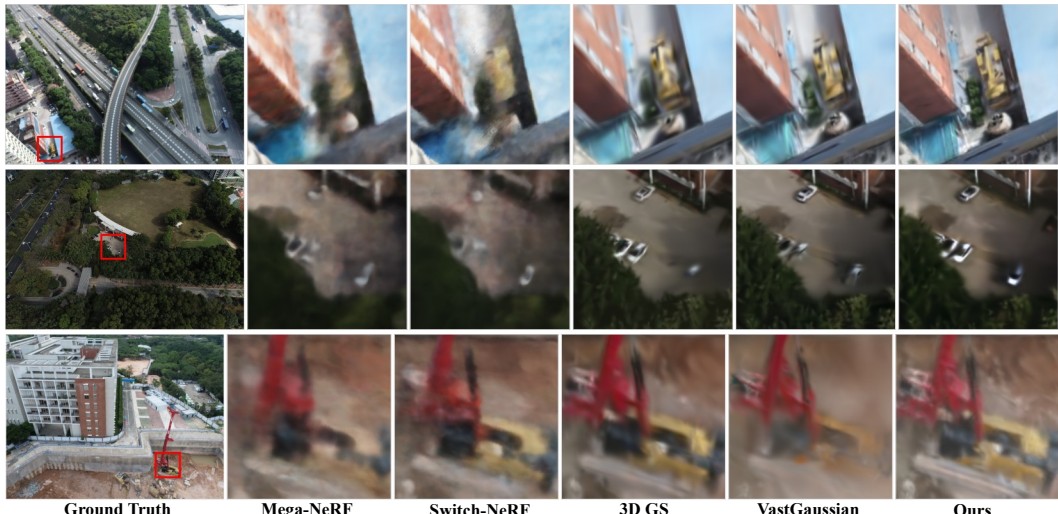

**Ground Truth**    **Mega-NeRF**    **Switch-NeRF**    **3D GS**    **VastGaussian**    **Ours**

Figure 5: **Qualitative comparisons of our method and others on the UrbanScene3D dataset**. From top to bottom are respectively the results of scenes 'campus', 'residence', and 'sci-art'.

Table 3: **Quantitative results of novel view synthesis on the MatrixCity [24] dataset**. ↑: higher is better, ↓: lower is better. The red, orange and yellow colors respectively denote the best, the second best, and the third best results.

| Scenes | aerial | | | | | | | street | | | | | | |
|---|---|---|---|---|---|---|---|---|---|---|---|---|---|---|
| | PSNR ↑ | SSIM ↑ | LPIPS ↓ | Time ↓ | Points | Mem ↓ | FPS ↑ | PSNR ↑ | SSIM ↑ | LPIPS ↓ | Time ↓ | Points | Mem ↓ | FPS ↑ |
| 3DGS | 27.36 | 0.818 | 0.237 | 47:40 | 11.9 | 6.31 | 45.57 | 20.03 | 0.643 | 0.650 | 14:24 | 1.85 | 2.33 | 193.32 |
| VastGaussian[†] | 28.33 | 0.835 | 0.220 | 05:53 | 12.5 | 6.99 | 40.04 | - | - | - | - | - | - | - |
| DOGS | 28.58 | 0.847 | 0.219 | 06:34 | 10.3 | 5.82 | 48.34 | 21.61 | 0.652 | 0.649 | 02:33 | 2.37 | 2.89 | 180.51 |

the over-relaxation is denoted as *w.o. OR*. As shown in the table, the performance drastically drops without the ADMM consensus step. Furthermore, the results without applying the self-adaptation of penalty parameters is about 1.5 dB lower than the full model in PSNR. The model without applying over-relaxation is comparable to the full model in SSIM and LPIPS but has lower PSNR. We thus employ over-relaxation in our method. We also present the qualitative differences in Fig. 7, and it clearly shows the full model has better quality in the rendered images and geometries. We include more qualitative results in Fig. 8 to show the importance of the consensus step. We can observe that the distributed training presents noisy results without the consensus step. From the two bottom-right figures, we can observe obvious artifacts along the block boundary without the consensus step.

Moreover, we ablate the scale factor in constructing the overlapping areas in Table 5. We can find that the performance of our method is improved with a larger scale factor. Our method has similar performance when the scale factor is $1.4$ and $1.5$. However, we select $1.4$ in our experiments since a larger scale factor brings a longer time and more memory requirement.

## 5    Conclusion

In this paper, we proposed DOGS, a scalable method for training 3DGS distributedly under large-scale scenes. Our method first splits scenes into multiple intersected blocks recursively and then trains each block distributedly on different workers. Our method additionally maintains a global 3DGS model on the master node. The global 3DGS model is shared with each block to encourage each block to converge to the global model. The local 3DGS of all blocks is collected to update the global 3DGS model. When evaluated on large-scale datasets, our method accelerates the 3DGS training time by $6\times \sim 8\times$ while achieving the best rendering quality in novel view synthesis.

Table 4: **Ablation study** of our method.

| | PSNR ↑ | SSIM ↑ | LPIPS ↓ |
|---|---|---|---|
| w.o. CS | 22.80 | 0.677 | 0.326 |
| w.o. SD | 24.30 | 0.729 | 0.285 |
| w.o. OR | 24.45 | 0.766 | 0.259 |
| full model | 25.78 | 0.765 | 0.257 |

Table 5: **Ablation study** of the scale factor in our method.

| | PSNR ↑ | SSIM ↑ | LPIPS ↓ | Times ↓ | Points | FPS ↑ |
|---|---|---|---|---|---|---|
| 1.2 | 24.25 | 0.739 | 0.276 | 02:27 | 4.95 | 129.87 |
| 1.3 | 24.86 | 0.750 | 0.270 | 02:27 | 5.02 | 128.73 |
| 1.4 | 25.78 | 0.765 | 0.257 | 02:25 | 4.74 | 147.06 |
| 1.5 | 25.97 | 0.767 | 0.257 | 02:39 | 4.84 | 130.28 |

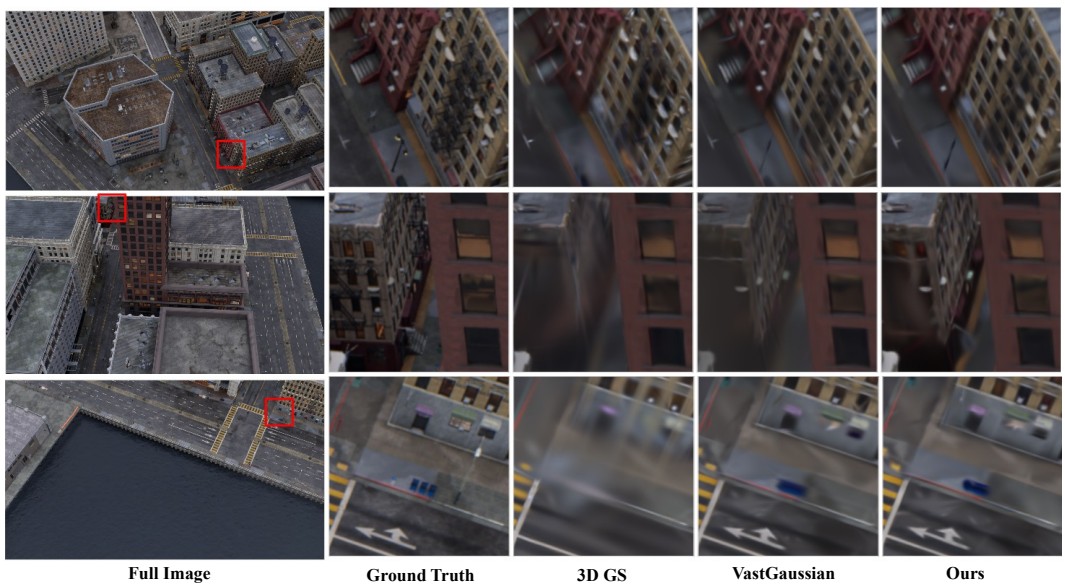

| Full Image | Ground Truth | 3D GS | VastGaussian | Ours |

Figure 6: Qualitative results on the MatrixCity [24] dataset.

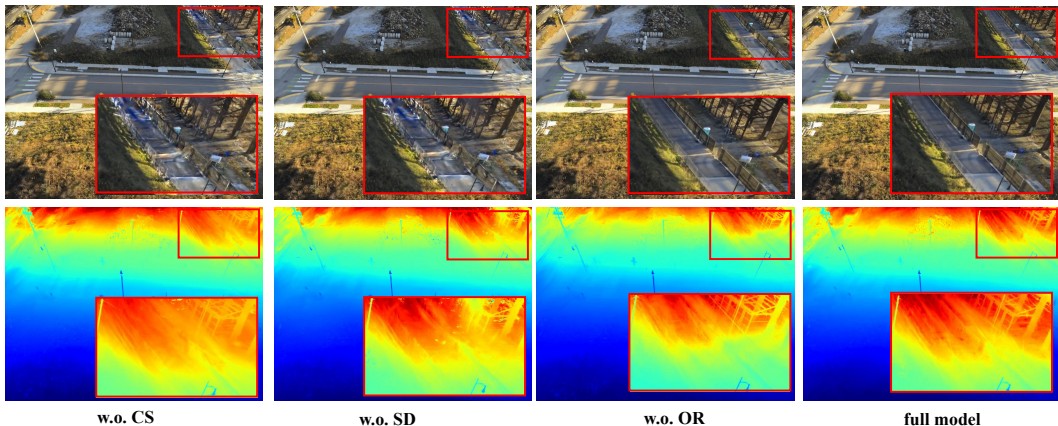

| w.o. CS | w.o. SD | w.o. OR | full model |

Figure 7: **Ablation study** of our method. Top: rendered images; Bottom: rendered depths.

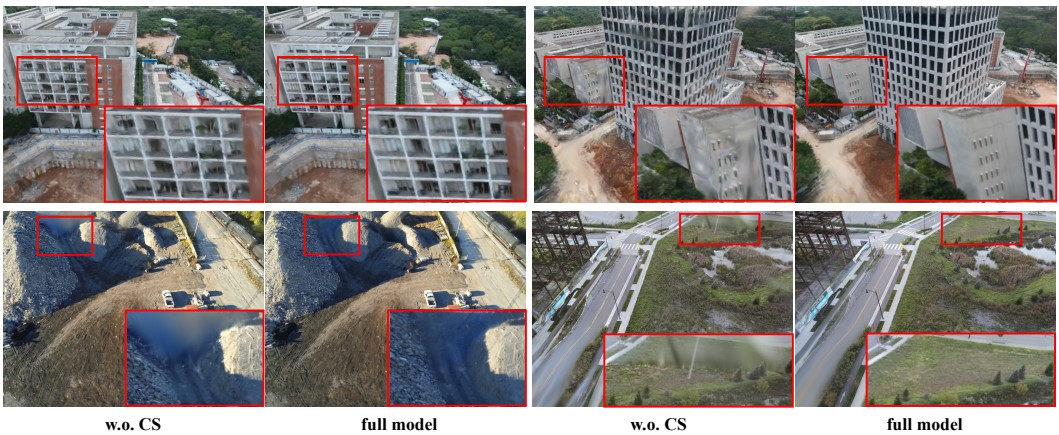

| w.o. CS | full model | w.o. CS | full model |

Figure 8: Importance of the consensus step.

**Acknowledgement.** This research / project is supported by the National Research Foundation (NRF) Singapore, under its NRF-Investigatorship Programme (Award ID. NRF-NRFI09-0008). Yu Chen is also partially supported by a Google PhD Fellowship.

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

# A Appendix

## A.1 Derivation of Eq. (7)

Given Eq. (5) $\mathcal{L}_\rho(\mathbf{x}, \mathbf{z}, \mathbf{y}) = \sum_{i=1} \left( f_i(\mathbf{x}_i) + \mathbf{y}_i^\top (\mathbf{x}_i - \mathbf{z}) + \frac{\rho}{2} \|\mathbf{x}_i - \mathbf{z}\|_2^2 \right)$, $\mathbf{r}_i = \mathbf{x}_i - \mathbf{z}$ and $\mathbf{u}_i = \frac{1}{\rho} \mathbf{y}_i$, we have

$$
\begin{aligned}
& \mathbf{y}_i^\top (\mathbf{x}_i - \mathbf{z}) + \frac{\rho}{2} \|\mathbf{x}_i - \mathbf{z}\|_2^2 \\
&= \mathbf{y}_i^\top \mathbf{r}_i + \frac{\rho}{2} \|\mathbf{r}_i\|_2^2 \\
&= \frac{\rho}{2} \|\mathbf{r}_i + \frac{\rho}{2} \mathbf{y}_i\|_2^2 - \frac{1}{2\rho} \|\mathbf{y}_i\|_2^2 \\
&= \frac{\rho}{2} \|\mathbf{r}_i + \mathbf{u}_i\|_2^2 - \frac{\rho}{2} \|\mathbf{u}_i\|_2^2.
\end{aligned}
\tag{11}
$$

Substitute Eq. (11) into Eq. (5), we have the scaled form ADMM:

$$
\mathcal{L}_\rho(\mathbf{x}, \mathbf{z}, \mathbf{u}) = \sum_{i=1}^N \left( f_i(\mathbf{x}_i) + \frac{\rho}{2} \|\mathbf{x}_i - \mathbf{z} + \mathbf{u}_i\|_2^2 - \frac{\rho}{2} \|\mathbf{u}_i\|_2^2 \right)
$$

## A.2 Proof of Eq. (8b)

The complete form of Eq. (8b) of is:

$$
\mathbf{z}^{t+1} = \frac{1}{K} \sum_{i=1} \left( \mathbf{x}_i^{t+1} + \mathbf{u}_i^t \right).
\tag{12}
$$

We denote the average of a vector with an overline. Then the consensus step for Eq. (8b) can be rewritten by:

$$
\mathbf{z}^{t+1} = \bar{\mathbf{x}}^{t+1} + \bar{\mathbf{u}}^t.
\tag{13}
$$

Moreover, averaging both sides of Eq. (8c) for the dual variables gives:

$$
\bar{\mathbf{u}}^{t+1} = \bar{\mathbf{u}}^t + \bar{\mathbf{x}}^{t+1} - \mathbf{z}^{t+1}.
\tag{14}
$$

Substituting Eq. (13) into Eq. (14) gives that $\bar{\mathbf{u}}^{t+1} = \mathbf{0}$, which shows that the dual variables have average value zeros after the first iteration. Therefore, we proved $\mathbf{z}^{t+1} := \frac{1}{K} \sum_{i=1} \left( \mathbf{x}_i^{t+1} \right)$.

## A.3 Detailed form of Eq. (8)

We provide the detailed form of Eq. (8) in this section. Note that each 3D Gaussian can be represented by $\mathbf{X}_i = \{\mathbf{u}_i, \mathbf{q}_i, \mathbf{s}_i, \mathbf{f}_i, o_i\}$, we need to apply different penalty terms to different properties of 3D GS. Specifically, the dual variables correspond to different properties of 3D GS are denoted as:

$$
\mathbf{u}_i = \{\mathbf{u}_p, \mathbf{u}_q, \mathbf{u}_s, \mathbf{u}_f, \mathbf{u}_o\},
\tag{15}
$$

where $\mathbf{u}_p, \mathbf{u}_q, \mathbf{u}_s, \mathbf{u}_f, \mathbf{u}_o$ are respectively denote the dual variable corresponds to the mean $\mathbf{p}$, the quaternion $\mathbf{u}_q$, the scaling matrix $\mathbf{u}_s$, the feature vectors to encode color information $\mathbf{u}_f$, and the

opacity $\mathbf{u}_o$. Therefore, we can expand Eq. (8) as:

$$(\mathbf{x}_i^k)^{t+1} := \arg\min \left( f(\mathbf{x}_i^k) + \frac{\rho_p}{2} \|(\mathbf{p}_i^k)^t - \mathbf{z}_p^t + (\mathbf{u}_{p,i})^t\|_2^2 + \frac{\rho_q}{2} \|(\mathbf{q}_i^k)^t - \mathbf{z}_q^t + (\mathbf{u}_{q,i}^k)^t\|_2^2 \right.$$

$$+ \frac{\rho_s}{2} \|(\mathbf{s}_i^k)^t - \mathbf{z}_s^t + (\mathbf{u}_{s,i}^k)^t\|_2^2 + \frac{\rho_f}{2} \|(\mathbf{f}_i^k)^t - \mathbf{z}_f^t + (\mathbf{u}_{f,i}^k)^t\|_2^2$$

$$+ \frac{\rho_o}{2} \|(\mathbf{o}_i^k)^t - \mathbf{z}_o^t + (\mathbf{u}_{o,i}^k)^t\|_2^2), \tag{16a}$$

$$\mathbf{z}_p^{t+1} := \frac{1}{K} \sum_{k=1}^K (\mathbf{p}_i^k)^{t+1}, \ \mathbf{z}_q^{t+1} := \frac{1}{K} \sum_{k=1}^K (\mathbf{q}_i^k)^{t+1}, \ \mathbf{z}_s^{t+1} := \frac{1}{K} \sum_{k=1}^K (\mathbf{s}_i^k)^{t+1},$$

$$\mathbf{z}_f^{t+1} := \frac{1}{K} \sum_{k=1}^K (\mathbf{f}_i^k)^{t+1}, \mathbf{z}_o^{t+1} := \frac{1}{K} \sum_{k=1}^K (\mathbf{o}_i^k)^{t+1}, \tag{16b}$$

$$(\mathbf{u}_{p,i}^k)^{t+1} := (\mathbf{u}_{p,i}^k)^t + (\mathbf{p}_i^k)^{t+1} - \mathbf{z}^{t+1}, \ (\mathbf{u}_{q,i}^k)^{t+1} := (\mathbf{u}_{q,i}^k)^t + (\mathbf{q}_i^k)^{t+1} - \mathbf{z}^{t+1},$$

$$(\mathbf{u}_{s,i}^k)^{t+1} := (\mathbf{u}_{s,i}^k)^t + (\mathbf{s}_i^k)^{t+1} - \mathbf{z}^{t+1}, \ (\mathbf{u}_{f,i}^k)^{t+1} := (\mathbf{u}_{f,i}^k)^t + (\mathbf{f}_i^k)^{t+1} - \mathbf{z}^{t+1},$$

$$(\mathbf{u}_{o,i}^k)^{t+1} := (\mathbf{u}_{o,i}^k)^t + (\mathbf{o}_i^k)^{t+1} - \mathbf{z}^{t+1}. \tag{16c}$$

### A.4 Algorithm for Recursive Scene Splitting

We present the algorithm for the recursive scene splitting in Alg. 1.

---
**Algorithm 1** Recursive Scene Splitting Algorithm

---
**Input:** 3D points $\{\mathbf{X}_i\}$, number of blocks $K$.
**Output:** Local 3D points in each block $\mathcal{X} = \{\mathbf{X}_{i,k}\}$
 1: Estimate a bounding box aabb which tightly covers all 3D points.
 2: Initialize cells $\mathcal{C} = \{\text{aabb}\}$, local 3D points $\mathcal{X} = \emptyset$
 3: **while** $|\mathcal{C}| < K$ **do**
 4:      Let current cells $\mathcal{C}_{\text{cur}} \leftarrow \mathcal{C}$
 5:      **while** $|\mathcal{C}_{\text{cur}}| > 0$ **do**
 6:          Pop a bounding box in current cells aabb $:= \mathcal{C}_{\text{cur}}.\text{pop}(0)$
 7:          Remove the bounding box from cells $\mathcal{C} := \mathcal{C} - \{\text{aabb}\}$
 8:          Split the bounding box into two sub-cells $\text{aabb}_1$, $\text{aabb}_2$ along the longer axis
 9:          Group points into two blocks $\{\mathbf{X}_1, \mathbf{X}_2\}$
10:          Re-estimate two tighter bounding boxes $\text{aabb}_1'$, $\text{aabb}_2'$ for $\{\mathbf{X}_1, \mathbf{X}_2\}$
11:          Push bounding boxes $\text{aabb}_1'$, $\text{aabb}_2'$ into cells: $\mathcal{C} := \mathcal{C} + \{\text{aabb}_1', \text{aabb}_2'\}$
12: **for** block $k \in |\mathcal{C}|$ **do**
13:      Group points $\mathcal{X}_k$ located in the same cell $\mathcal{C}_k$ into a same block
14:      Update $\mathcal{X}$ by $\mathcal{X} := \mathcal{X} + \mathcal{X}_k$

---

At line 5, $|\cdot|$ denotes the capacity of a set. Algorithm 1 is adopted to split both the training views and sparse point clouds from SfM.

### A.5 Algorithm for Distributed Training of 3D GS

We present the algorithm for the distributed training of 3D Gaussian Splatting in Alg. 2.

Note that we adopt Eq. (16) in our implementation.

### A.6 More Implementation Details

All experiments are conducted on the Nvidia RTX 6000 GPUs with 48 GB memory. For our method, we initialize the dual variables $\mathbf{u}_p, \mathbf{u}_q, \mathbf{u}_s, \mathbf{u}_f, \mathbf{u}_o$ to zeros. For the penalty parameters, we set $\rho_p, \rho_q, \rho_s, \rho_o$ to $1e4$ and $\rho_f$ to $1e3$ empirically. Though there are other choices of the initial values

**Algorithm 2** Distributed 3D Gaussian Training Algorithm

---

**Input:** Initial 3D Gaussians in each block $\{\mathbf{X}_k | \, k \in [1, K]\}$, consensus interval intv
**Output:** Global 3D Gaussians $\mathbf{z} = \{\mathbf{u}_i, \ \mathbf{q}_i, \mathbf{s}_i, \mathbf{f}_i, o_i\}$
 1: Initialize $\mathbf{u}_i$ as 0
 2: **for** $t < T$ **do**
 3:     **for** block $k < K$ distributedly **do**
 4:         $\{\mathbf{u}_{i,k}^{t+1}, \ \mathbf{q}_{i,k}^{t+1}, \mathbf{s}_{i,k}^{t+1}, \mathbf{f}_{i,k}^{t+1}, o_{i,k}^{t+1}\} := \arg\min \left( f_i(\mathbf{x}_{i,k}) + \frac{\rho}{2} \|\mathbf{x}_{i,k}^t - \mathbf{z}^t + \mathbf{u}_{i,k}^t\|_2^2 \right)$
 5:     **if** $t$ mod intv $== 0$ **then**
 6:         **for** block $k < K$ distributedly **do**
 7:             Send local 3D Gaussians $\mathbf{X}_k^{t+1}$ to the master node
 8:         Apply the consensus step $\mathbf{z}^{t+1} := \frac{1}{K} \sum_{i=1}^{K} \left( \mathbf{x}_{i,k}^{t+1} \right)$
 9:         Broadcast the global 3D Gaussians $\mathbf{z}^{t+1}$ to all slave nodes
10:         **for** block $k < K$ distributedly **do**
11:             Update the dual variables $\mathbf{u}_{i,k}^{t+1} := \mathbf{u}_{i,k}^t + \mathbf{x}_{i,k}^{t+1} - \mathbf{z}^{t+1}$

---

that could improve the results of our method, we found this set of values is good enough for all scenes in our experiments and we did not do more ablations on the initial values. Due to computational resources limitation, we test our method on only 5 GPU nodes, where one is the master node that maintains the global model and the others are slave nodes for training local 3D Gaussians. The performance of our method can be improved further with more GPU nodes.

### A.7 More Qualitative Results

We present more qualitative results of our method in Fig. 9 and Fif. 10. We provide qualitative comparisons with VastGaussian in areas where blocks overlap in Fig. 10. Both methods produce fairly consistent results. However, our method presents higher fidelity rendering results than VastGaussian near the splitting boundary, which also validated the effectiveness and importance of the consensus step.

### A.8 Limitations and Future Work

Our method can distributedly train 3D GS on large-scale scenes. However, it brings additional communication overheads to the system. Fortunately, we found the communication overhead did not slow down the training performance. This is due to our balanced splitting algorithm minimizing the synchronization time when there is a need to consensus the local 3D Gaussians from all blocks. Moreover, we pruned unnecessary small 3D Gaussians to further reduce the number of 3D GS, which also reduced the communication overhead.

Though our method can train large-scale scenes efficiently, the GPU memory requirement can still be an issue. This is because when we zoom out to capture larger areas, more 3D Gaussians are included in the rasterization step, which consumes more GPU memory. Our future work will consider introducing the level-of-details (LOD) technique into our distributed training pipeline. Similar to existing LOD GS methods [41, 44], LOD can be utilized to reduce the number of 3D Gaussians that are far away from the cameras.

### A.9 More Discussions

**Training and Waiting Time of Each Block.** We tested the time cost from transferring the data to the master node to receiving data from the master node for each slave node on the Campus dataset. The mean and variance of time are respectively 5.63 seconds and 0.75 seconds each time. The low variance indicates that our method can balance the training time well. We argue that the data transfer time of our method can be kept constant since we can always control the number of local Gaussians to a constant number (*e.g.* $<= 6000,000$ 3D Gaussian primitives) with enough GPUs, no matter how large the scene is, since the data transfer between different slave nodes and the master node is executed distributedly instead of sequentially.

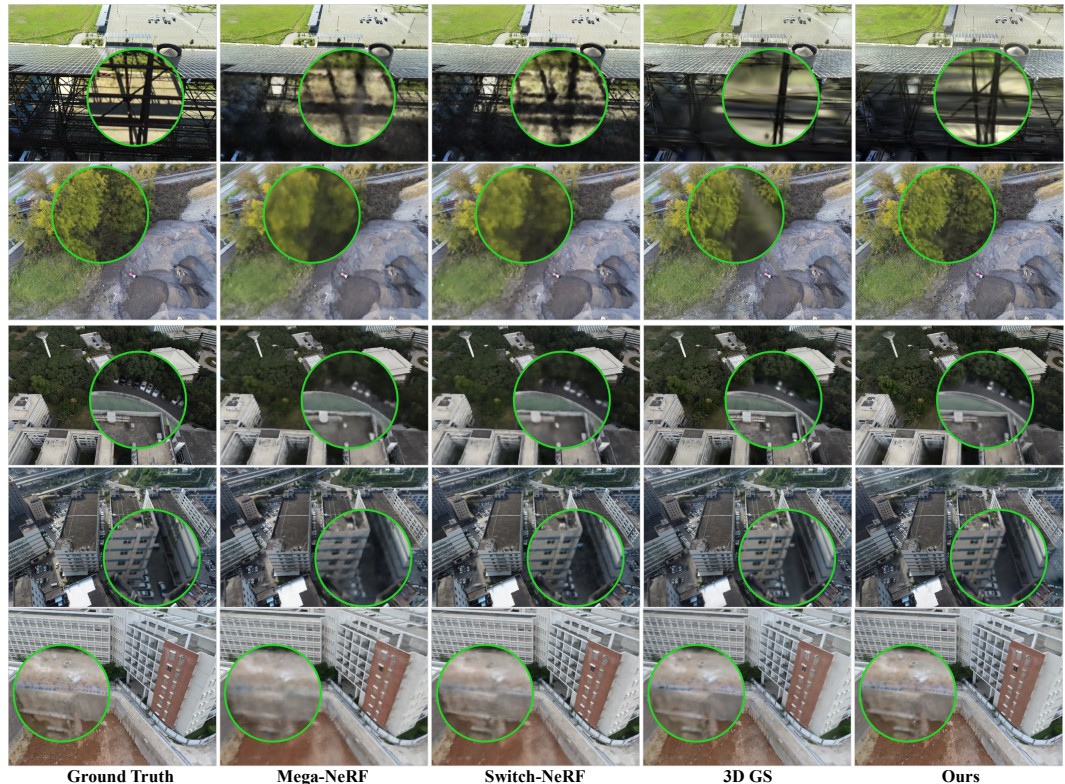

| Ground Truth | Mega-NeRF | Switch-NeRF | 3D GS | Ours |
| --- | --- | --- | --- | --- |

Figure 9: More qualitative results on the mill19 dataset and the UrbanScene3D dataset. Zoom in for the best view.

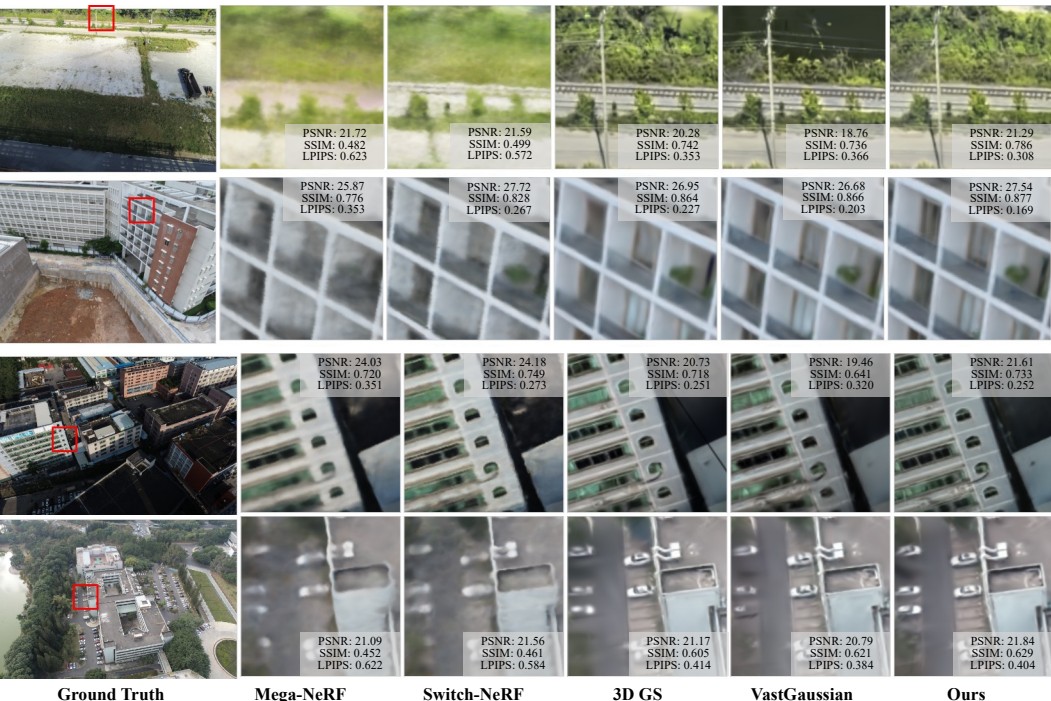

| Ground Truth | Mega-NeRF | Switch-NeRF | 3D GS | VastGaussian | Ours |
| --- | --- | --- | --- | --- | --- |

Figure 10: More qualitative comparisons of our method and state-of-the-art methods.

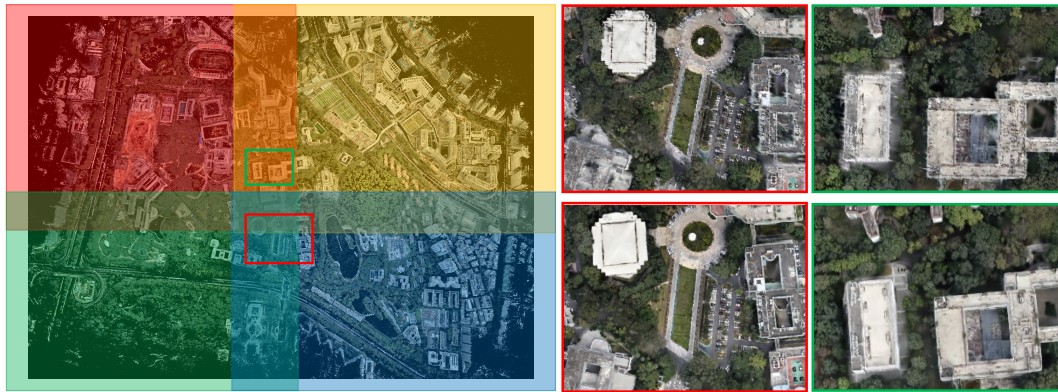

Figure 11: Qualitative results near scene boundary on the UrbanScene3D dataset. Top-Right: VastGaussian. Bottom-Right: our method.

**Implementation Flexibility.** One another issue with our method is that the implementation of our method is not as flexible as VastGaussian. Our method relies on an RPC module for the data transmission and communication between the master node and the slave nodes. On the other hand, VastGaussian can be implemented decentralized without the master node. However, our method can render higher-quality images and the communication overheads can be neglected compared to the long training time of the original 3D GS. Therefore, this is not a limitation of our work.

**Shift to Other 3D GS Representations.** Many follow-up works improved the original 3D GS. Some works focus on compressing the size of 3D GS [16, 38] and some of the other works focus on constraining the training without changing the optimization parameters [64, 23, 13]. These methods can also be applied to the training of each block in our method. Some other works also adopted intermediate representations to improve the original 3D GS, *i.e.*, OctreeGS [41] decoded the properties of 3D GS from the feature embedding in each anchor node, and SAGS [53] adopts hash encodings for each 3D Gaussian, a GNN encoder, and a corresponding decoder to generate the properties of 3D Gaussians. In this case, we can change the optimization parameters and the corresponding penalty parameters and dual variables to these intermediate feature embeddings/encodings. In our future work, we will consider a more consolidated implementation that can easily shift to these 3D GS representations.

**Comparison to Concurrent Works.** 1) **VastGaussian** [26] focuses mostly on the data splitting strategy and guaranteeing the consistency of different 3D Gaussians can also be a challenge in different scenes. However, our method ensured the consistency of the shared 3D Gaussians through the 3D Gaussian consensus with only a quite simple splitting approach. Nonetheless, the data splitting approach in VastGaussian can be introduced into our framework to enhance the robustness. Moreover, the decoupled appearance encoding can also be applied to our method to reduce floaters. Our work and VastGaussian are thus complementary to each other. 2) **Hierarchy-GS** [22] also trains 3DGS distributedly in different grid cells. However, Hierarchy-GS focuses more on the rendering speed by designing a hierarchical tree structure for 3D Gaussians. During training, the generated hierarchy is loaded and optimized by randomly selecting a training view and a target granularity. After training all individual chunks, Hierarchy-GS first runs a coarse initial optimization on the full dataset and adds an auxiliary skybox. This coarse model serves as a minimal basis for providing backdrop details for parts of the scene that are outside of a chunk. When consolidating different chunks, a 3D Gaussian primitive is deleted if it is associated to-but outside-chunk and is closer to another chunk to maintain the consistency of 3D Gaussians. The naive pruning approach neglects the gradient flow of the 3D Gaussians to other 3D Gaussians that are inside only in an associated chunk. Therefore, its rendering quality can be worse than VastGaussian and our method. Nonetheless, its hierarchical LOD approach can still serve as a good complement to our method and VastGaussian.

**Social Impact.** DOGS, VastGaussian and Hierarchy-GS focus on different parts of training 3D GS on large-scale scenes and therefore are complementary to each other. These methods can be consolidated into a more robust and scalable system to open a new world in city-scale 3D reconstruction. Our method, however, requires an additional master node for controlling the training of the 3DGS, which consumes more GPU resources.

