# OpenReview forum: "DOGS: Distributed-Oriented Gaussian Splatting for Large-Scale 3D Reconstruction  Via Gaussian Consensus"
_NeurIPS.cc/2024/Conference — NeurIPS 2024 poster_

### Official Review · Reviewer_eg7b · 2024-06-17

**Soundness:** 4
**Presentation:** 4
**Contribution:** 3
**Rating:** 6
**Confidence:** 5

**Summary:**

DoGaussian incorporates the 'divide-and-conquer' approach and introduces the ADMM algorithm into the 3DGS training process for large-scale 3D reconstruction tasks, reducing training time by 6+ times compared to the original 3DGS. Specifically, DoGaussian first splits the scenes into K local blocks of similar sizes and then maintains a global 3DGS node to ensure consistency through consensus on the K shared blocks. During the inference stage, DoGaussian moves all shared blocks and queries only the global block to improve rendering efficiency.

**Strengths:**

1. DoGaussian proposes a novel distributed training strategy, achieving a significant acceleration of the 3DGS training process without sacrificing rendering visual quality.
2. DoGaussian functions more like a plugin and can be applied to any GS representation work.
3. The paper is well-written and easy to follow, and the supplementary video is exceptionally well made.

**Weaknesses:**

1. **Artifacts in Teaser.** There are noticeable artifacts in the teaser, particularly in the picture located in the bottom right corner. It would be beneficial for the authors to provide an explanation of these artifacts and address them accordingly.
2. **Suitable for Street Scenes.** In the experiments section, all experiments are conducted on aerial datasets, but reconstructing street data is also an important problem. Can the authors demonstrate DoGaussian's applicability using the San Francisco Mission Bay dataset from Block-NeRF[1] or Block_A from the MatrixCity[2] dataset?
3. **GPU Memory Problem.** DoGaussian appears to be a system paper that utilizes a distributed approach to improve training efficiency by placing the global 3DGS on a main node and some local 3DGS on other nodes. One potential reason other works do not use this strategy could be GPU memory limitations. All datasets used by the authors are processed by 3DGS on a single GPU, which does not truly qualify as 'large-scale.' It would be beneficial for the authors to demonstrate that DoGaussian can be applied to larger datasets, such as the aerial and street data of big city in MatrixCity, which is used in NeRF-XL[3].

[1] Block-nerf: Scalable large scene neural view synthesis.

[2] Matrixcity: A large-scale city dataset for city-scale neural rendering and beyond

[3] NeRF-XL: Scaling NeRFs with Multiple GPUs

**Questions:**

There are three questions regarding the weaknesses mentioned above:

1. The authors need to explain the artifacts in the teaser.
2. It would be better to conduct experiments on street datasets and larger datasets.
3. I’m also curious to know if the final results would be better using the visibility-aware splitting strategy of VastGaussian.

I would be happy to raise my score if the authors can address all my questions.

**Limitations:**

The authors discuss the limitation in the appendix A.8.

---

> ### Author Rebuttal · Authors · 2024-08-06
>
> - ***Q1: Noticeable artifacts in the teaser, particularly in the picture located in the bottom right corner.***
> - **A1:** From Fig.3 in the attached PDF, we can observe that the artifacts in the teaser appear to be near the scene boundary, which is a common problem for 3DGS-based methods and not a particular issue caused by our method.
>
> - ***Q2: Suitable for Street Scenes.***
> - **A2:**: See **A4** in the common questions.
>
> - ***Q3: GPU Memory Problem***
> - **A3:** See **A1**, **A2** and **A3** in the common questions.
>
> - ***Q4: It would be better to conduct experiments on street datasets and larger datasets, such as the aerial and street data of Big City in MatrixCity, which is used in NeRF-XL.***
> - **A4:** Since we have only a limited number of GPUs with 48GB memory for each, we did not test our method on the Big City scene in MatrixCity. However, to further show the applicability of our method to larger-scale scenes, we evaluated our method on the $2.7 \text{km}^2$ Small City scene in the MatrixCity dataset, which contains $5,620$ training views and $741$ validation views. We early terminated the training of the original 3D-GS since it did not finish the training within two days. From the table below, our method achieved the best results in rendering quality. The visual qualitative results are shown in Fig.5 in the attached PDF.
>
> *Table. Quantitative results of our method on the Small City aerial scene of the MatrixCity dataset.*
> |            | PSNR $\uparrow$ | SSIM $\uparrow$ | LPIPS $\downarrow$ | Time $\downarrow$ | Points | Mem $\downarrow$ | FPS $\uparrow$ |
> |------------|-----------------|--------------|-----------------|--------------|--------|--------------|-------------|
> | 3D-GS      | 27.36 | 0.818 | 0.237 | 47:40 | 11.9 | 6.31 | 45.57 |
> | VastGaussian$^{\dagger}$ | 28.33 | 0.835 | 0.220 | 05:53 | 12.5 | 6.99 | 40.04 |
> | DoGaussian | **28.58** | **0.847** | **0.219** | 06:34 | 10.3 | 5.82 | 48.34 |
>
> - ***Q5: If the final results would be better using the visibility-aware splitting strategy of VastGaussian.***
> - **A5:** The visibility-aware splitting strategy of VastGaussian is utilized to add more redundant training views to each block to ensure well-constrained scene boundaries. However, this strategy also imbalances the size of each block. In our experiments, we find this strategy improves little of our method in the rendered image quality but increases the training time. On the Campus scene, when adopting the visibility-aware strategy in our method, the PSNR is increased by 0.09db, but the training time is increased by 25 minutes. While visibility-awareness is a good strategy for improving the rendering quality of the model, some post-processing steps may be required to ensure balanced block-splitting results, which can be a future work of our method.

---

> > ### Comment · Reviewer_eg7b · 2024-08-08
> >
> > Thanks for your reply and effort. All my concerns have been addressed. I keep my original score.

---

> > > ### Author Response · Authors · 2024-08-10
> > > **Thanks for your reply**
> > >
> > > We are grateful for your insightful suggestions and happy our rebuttal has addressed your concerns. We will update the paper with all the experiments and details mentioned.

---

### Official Review · Reviewer_tFmz · 2024-07-11

**Soundness:** 3
**Presentation:** 3
**Contribution:** 3
**Rating:** 6
**Confidence:** 3

**Summary:**

This paper proposes a distributed training strategy for 3dgs in large-scale scenes. The scene is evenly splitted into K blocks, but also maintain the global scene representation. Then the optimization of the scene is transferred to a condition optimization solved by the classic   ADMM. The results demonstrate their effectiveness and efficiency.

**Strengths:**

1. It is interesting to convert the distributed training of 3dgs into a condition optimization problem. It could boost the rendering in overlapping region significantly.

2. The experiments are sufficient.

**Weaknesses:**

1. The scene splitting is somehow tailored for bird view images. As for the extending scenes with surround views, such as autonomous driving, a training image in one block could see a large ratio of point cloud in other blocks.

2. It would be better to modify some representations in the paper.

a. line 82, cone sampling of mip-nerf is not designed for the representation enhancement for outdoor scenes.

b. line 98, "leveraging xxx performance for xxx reconstruction" seems confusing.

c. line 4 of the fig.2, there are a redundant 'the' in sentence "a copy the the global 3D Gaussians".

3. It would be better to highlight the best performance in Table 2, 4.

**Questions:**

1. How to load the global gaussians? Does the method work when a single GPU can not load the global scene (main node)?

2. As the different splitting could lead to different overlapping regions and different ADMM problems, does it matter?

**Limitations:**

As discussed in the Question 1.

---

> ### Author Rebuttal · Authors · 2024-08-06
>
> - ***Q1: applicability on autonomous driving scenes***
> - **A1:** See **A4** in the common questions.
>
> - ***Q2: It would be better to modify some representations in the paper***
> - **A2:** Thanks for the suggestions. We revised the paper accordingly as suggested:
> ***(a)*** line 82, we revised it to "To address the aliasing issue in vanilla NeRF, Mip-NeRF[1] proposed to use Gaussian to approximate the cone sampling, ...".
> ***(b)*** line 98, we revised it as "or leveraging the efficient rasterizer of point rendering for real-time indoor reconstruction".
> ***(c)*** line 4 of the fig.2, we revised it as "a copy of the global 3D Gaussians".
>
> - ***Q3: It would be better to highlight the best performance in Table 2, 4.***
> - **A3**: Thanks for the suggestion. We will highlight the best results in Tables 2 and 4 in our revised paper as is done in Table 1, 3.
>
> - ***Q4: How to load the global gaussians? Does the method work when a single GPU can not load the global scene (main node)?***
> - **A4:** See **A2** in the common questions.
>
> - ***Q5: As the different splitting could lead to different overlapping regions and different ADMM problems, does it matter?***
> - **A5:** It is possible that better splitting methods can lead to better results and boost the convergence of the consensus ADMM problems. From Eq.8(b) of the main paper, the global Gaussians are updated by averaging the values from local Gaussians. Though local Gaussians converge towards the global Gaussians during training, a high variance of the shared local Gaussians (it occurs when Gaussians are not well constrained) can decrease the convergence of ADMM. Our method does not suffer from this issue since we adopt a confidential scale factor to ensure the scene boundaries have enough overlapping areas.

---

> ### Comment · Reviewer_tFmz · 2024-08-08
>
> Thanks for your reply and answers. Most of my concerns are addressed. Although the limitation in street scenes exists currently, the contribution of this paper for bringing the condition optimization into large scale scene learning is clear. So I keep my original score.

---

> > ### Author Response · Authors · 2024-08-10
> > **Thanks for your reply**
> >
> > We are very grateful for your insightful suggestions and support of our work. We are happy to see that our rebuttal has addressed most of your concerns.

---

### Official Review · Reviewer_zf9w · 2024-07-12

**Soundness:** 3
**Presentation:** 3
**Contribution:** 3
**Rating:** 6
**Confidence:** 5

**Summary:**

This paper introduces traditional ADMM to Gaussian Spaltting and achieves distributed Gaussian Splatting training. The proposed distributed approach reduces training time and guarantes training convergence and stability. Experiments demonstrates both effectivenss and efficiency of this method.

**Strengths:**

1. The paper introduces powerful ADMM to large-scale 3D reconstruction, which is an effective and elegant approach to distributed optimization.
2. The compact optimization scheme help the method achieves high quality reconstruction.
3. The idea of the proposed distributed optimization could inspire research in other tasks, such as full-body reconstruction.
4. The paper is well written.

**Weaknesses:**

1. More experiments are needed to prove the superiority of the consensus step. I hope to see more qualitative and quantitative comparisons with VastGaussian in areas where blocks overlap.
2. The authors claim that proposed split method can balance the training time of each block. However, there is a lack of experiments showing the training and waiting time of each block.
3. The authors enable consensus and sharing when every block reaches 100 iterations. Is it possible to allow the blocks to be at different iterations? It can be more efficient to avoid blocking. However, there is no discussion about the difference between synchronous and asynchronous methods.
4. There is no discussion about the memory consumption of master node. How much GPU memory is needed for the master node? Will its memory consumption be linearly increase as the total number of Gaussians increases?

**Questions:**

NA

---

> ### Author Rebuttal · Authors · 2024-08-06
>
> - **Q1: More experiments are needed to prove the superiority of the consensus step. More qualitative and quantitative comparisons with VastGaussian in areas where blocks overlap.**
> - **A1:** We include more qualitative results in Fig.2 in the attached PDF to show the importance of the consensus step. We can observe that the distributed training presents noisy results without the consensus step. From the two bottom-right figures, we can observe obvious artifacts along the block boundary without the consensus step.
> As is shown in Fig.4 of the attached file, both methods produce fairly consistent results. However, our method presents higher fidelity rendering results than VastGaussian near the splitting boundary, which also validated the effectiveness and importance of the consensus step.
>
> - ***Q2: The authors claim that proposed split method can balance the training time of each block. However, there is a lack of experiments showing the training and waiting time of each block.***
> - **A2:** We tested the time cost from transferring the data to the master node to receiving data from the master node for each slave node on the Campus dataset. The mean and variance of time are respectively $5.63$ seconds and $0.75$ seconds each time. The low variance shows that our method can balance the training time very well.
>
> - ***Q3: The authors enable consensus and sharing when every block reaches 100 iterations. Is it possible to allow the blocks to be at different iterations? It can be more efficient to avoid blocking. However, there is no discussion about the difference between synchronous and asynchronous methods.***
> - **A3:** The consensus and sharing steps can be done asynchronously in implementation. However, by doing so, the convergence of ADMM is not guaranteed and the results can be sub-optimal. To validate this, (1) we first transfer the local Gaussians to the master node and then optimize them without blocking; (2) we trigger the consensus and sharing step once the master node receives local Gaussians from all blocks; (3) the local Guassians are regularized with the newly updated global Gaussians once the slave nodes received the data. As a result, the PSNR, SSIM, LPIPS on the building scene are respectively $18.73, 0.441, 0.673$, which dropped significantly.
>     We argue that **the data transfer time of our method can be kept constant** since we can always control the number of local Gaussians to a constant number (e.g. $<= 6000,000$ 3D Gaussian primitives) with enough GPUs, no matter how large the scene is, since the data transfer between different slave nodes and the master node is executed distributedly instead of sequentially.
>
> - ***Q4: memory consumption of master node***
> - **A4:** See **A1** in the common questions.

---

> > ### Comment · Reviewer_zf9w · 2024-08-11
> >
> > Thanks for your reply. I will keep my rating.

---

> > > ### Author Response · Authors · 2024-08-13
> > > **Thanks for your reply**
> > >
> > > Thanks for your reply and support of our work. We are very grateful for your insightful suggestions and will revise our paper accordingly.

---

### Official Review · Reviewer_Y2Rh · 2024-07-12

**Soundness:** 3
**Presentation:** 3
**Contribution:** 3
**Rating:** 6
**Confidence:** 2

**Summary:**

The paper presents DoGaussian, a novel method for accelerating the training of 3D Gaussian Splatting models for large-scale 3D scene reconstruction. It introduces a distributed training approach using scene decomposition and the ADMM, resulting in a 6x faster training time while maintaining high-quality rendering. The method involves maintaining a global 3DGS model on a master node and local models on slave nodes, with a consensus mechanism ensuring consistency across models. Experiments on standard large-scale datasets confirm the effectiveness of DoGaussian.

**Strengths:**

1. The use of the ADMM ensures that the global and local models reach a consensus on the shared 3D Gaussians. This guarantees convergence and stability during the training phase, which is crucial for maintaining the integrity of the model.
2. Despite the focus on efficiency, the paper does not compromise on the quality of the output. It demonstrates state-of-the-art rendering quality, which is a testament to the robustness of the proposed method.
3. By introducing a distributed training methodology for 3DGS, the paper significantly accelerates the training process on large-scale scenes. This is achieved by decomposing scenes into manageable blocks, allowing for parallel processing on multiple nodes.

**Weaknesses:**

1. The qualitative comparisons are not that convincing, especially compared with the original 3DGS and vastGaussian. The authors are encouraged to provide more results in the rebuttal. I will raise the score accordingly.
2. The training and inference speed of DoGaussian is slightly slower than vastGaussian.
3. More baselines should be included, such as [Fed3DGS](https://arxiv.org/pdf/2403.11460).

**Questions:**

Can you provide more qualitative results and comparisons?

**Limitations:**

Yes

---

> ### Author Rebuttal · Authors · 2024-08-06
>
> - ***Q1: Provide more qualitative results in the rebuttal (compared with the original 3DGS and VastGaussian).***
> - **A1:** We include more qualitative results in Fig.1 in the attached PDF.
>
> - ***Q2: More baselines should be included, such as Fed3DGS.***
> - **A2:** We include the results of Fed3DGS in our updated Table. We can observe that **the rendering quality of Fed3DGS is far below 3D-GS-based methods**. Note that though Fed3DGS also maintains a global model in a central server, it requires optimizing the opacities and appearance of the global model with all local models, which is computationally inefficient. Moreover, not like our method, which adopts ADMM to ensure the training convergence, the global model optimization in Fed3DGS is only used to prune Gaussians to prevent the monotonically increased number of global Gaussians, which is intuitively designed without theoretical convergence guarantee.
>
> | **Scenes** | **Building** | **Building**  | **Building**  | **Rubble** | **Rubble** | **Rubble** | **Campus** | **Campus** | **Campus** | **Residence** | **Residence** | **Residence** | **Sci-Art** | **Sci-Art** | **Sci-Art** |
> |------------|-------------------------------|------------------------------|---------------------------------|----------------------------|----------------------------|-------------------------------|----------------------------|----------------------------|--------------------------------|-------------------------------|-------------------------------|--------------------------------|-----------------------------|-----------------------------|--------------------------------|
> |  | PSNR$\uparrow$ | SSIM $\uparrow$ | LPIPS $\downarrow$ | PSNR $\uparrow$ | SSIM $\uparrow$ | LPIPS $\downarrow$ | PSNR $\uparrow$ | SSIM $\uparrow$ | LPIPS $\downarrow$ | PSNR $\uparrow$ | SSIM $\uparrow$ | LPIPS $\downarrow$ | PSNR $\uparrow$ | SSIM $\uparrow$ | LPIPS $\downarrow$ |
> | Mega-NeRF  | 20.92                         | 0.547                        | 0.454                           | 24.06                      | 0.553                      | 0.508                         | 23.42                      | 0.537                      | 0.636                          |  22.08 | 0.628                         | 0.401                          | 25.60 | 0.770                         | 0.312                          |
> | Switch-NeRF| 21.54                         | 0.579                        | 0.397                           | 24.31                      | 0.562                      | 0.478                         | 23.62                      | 0.541                      | 0.616                          | 22.57 | 0.654                         | 0.352                          | 26.51 | 0.795 | 0.271                          |
> | $\text{3D-GS}$ | 22.53 | 0.738 | 0.214 | 25.51 | 0.725 | 0.316 | 23.67 | 0.688 | 0.347 | 22.36 | 0.745 | 0.247 | 24.13 |0.791 | 0.262 |
> | $\text{Fed3DGS}$ | **18.66** | **0.602** | **0.362** | **20.62** | **0.588** | **0.437** | **21.64** | **0.635** | **0.436** | **20.00** | **0.665** | **0.344** | **21.03** | **0.730** | **0.335** |
> | $\text{VastGaussian}^{\dagger}$ | 21.80 | 0.728 | 0.225 | 25.20 | 0.742 | 0.264 | 23.82 | 0.695 | 0.329 | 21.01 | 0.699 | 0.261 | 22.64 | 0.761 | 0.261 |
> | Hierarchy-GS | 21.52 | 0.723 | 0.297 | 24.64 | 0.755 | 0.284 | -- | -- | -- | -- | -- | -- | -- | -- | -- |
> | DoGaussian | 22.73 | 0.759 | 0.204 | 25.78 | 0.765 | 0.257 | 24.01 | 0.681 | 0.377 | 21.94 | 0.740 | 0.244 | 24.42 | 0.804 | 0.219 |
>
> - ***Q3: More qualitative results***
> - **A3:** See the attached file for more qualitative results.

---

> > ### Comment · Reviewer_Y2Rh · 2024-08-13
> > **Thanks for your reply**
> >
> > Thanks for your reply. I will maintain my original score.

---

> > > ### Author Response · Authors · 2024-08-13
> > > **Thanks for your reply**
> > >
> > > Thanks for your rating. We are very grateful for your insightful suggestions and support of our work. We will update our paper accordingly as suggested in our revised version.

---

### Author Rebuttal · Authors · 2024-08-06

We sincerely thank all reviewers for their detailed comments and suggestions. Some **common questions are answered below**, and **more qualitative results** are provided in the attached PDF.
- ***Q1 How much GPU memory is needed for the master node? Will its memory consumption be linearly increase as the total number of Gaussians increases?***
- **A1**: In our experiments, the peak GPU memory of the master node is below 32 GB for all the datasets. The memory consumption increases linearly with the total number of Gaussians. However, we emphasize that **the global Guassians on the master node do not produce gradients** because we just use the value of local Gaussians in the consensus step to update the global Gaussians (we do not need the backpropagation for the optimizer to update the global Gaussians).
In implementation, we update the global Gaussians with torch's no gradient mode. We do not need to maintain a computational graph as we train the local Gaussians. Therefore, the memory consumption of maintaining the global Gaussians on the master node is much less than training the original 3DGS that has the same number of Gaussians on a single GPU.

- ***Q2: How to load the global gaussian? Does the method work when a single GPU cannot load the global scene (main node)?***
- **A2:** During training, the global Gaussians are maintained on the GPU of the master node. During consensus and sharing, local Gaussians can be mapped to their corresponding global Gaussians by indexing. During inference, the global Gaussians are used normally just as the original 3D-GS. We emphasize that **our method can still be applied to such scenes when a single GPU cannot load the global scene**: ***(1)*** As is explained in ***A1***, the memory requirement of the master node is not as large as training the same number of global Gaussians on a GPU. ***(2)*** In extremely large scenes where a single GPU cannot hold all global Gaussians (this is also an issue for all other large-scale 3DGS methods. For example, VastGaussian needs the final fused model to do inference), we can cache the global Gaussians on RAM and transfer them to GPU memory in a block-wise manner whenever the consensus and sharing step are triggered.
***In brief***, this is an implementation philosophy that can be solved as explained above and we will leave this as our future work.

- ***Q3: One potential reason other works do not use this strategy could be GPU memory limitations. All datasets used by the authors are processed by 3DGS on a single GPU, which does not truly qualify as 'large-scale.'***
- **A3:** The Mill19 and UrbanScene3D datasets are widely adopted in evaluating large-scale NeRF/3D-GS methods, and we follow previous methods (Mega-NeRF, Switch-NeRF, VastGaussian) to evaluate our method on these most commonly evaluated large-scale scenes. We emphasize that how we qualify a scene as `large-scale` is not simply by whether it can be trained on a single GPU (***e.g.*** if we have an 80GB GPU, we can train 3DGS on any existing datasets), but also on how long it takes to train a NeRF/3D-GS model on such scenes. We can train any scene on a single GPU by controlling the number of model parameters, but it can take a very long time for the model to converge to render high-fidelity images.

- ***Q4: In the experiments section, all experiments are conducted on aerial datasets, but reconstructing street data is also an important problem. Can the authors demonstrate DoGaussian's applicability using the San Francisco Mission Bay dataset from Block-NeRF or BlockA from the MatrixCity dataset?***
- **A4:** See the table below for the results on the BlockA scene of the MatrixCity dataset. VastGaussian failed on this dataset since two blocks produce no 3D Gaussian primitives due to the imbalanced splitting. Neither 3D-GS nor our method produces satisfactory results with the default parameters used to train 3D-GS, this may be because the BlockA scene contains too many narrow views and needs careful fine-tuning of the training parameters. However, our method still produces better results than the original 3D-GS. An exhaustive evaluation of our method on street view scenes can be a future work.

|          | PSNR $\uparrow$ | SSIM $\uparrow$ | LPIPS $\downarrow$ | Time $\downarrow$ | Points | Mem $\downarrow$ | FPS $\uparrow$ |
|------------|-----------------|--------------|-----------------|--------------|--------|--------------|-------------|
| 3D-GS      | 20.03 | 0.643 | 0.650 | 14:24 | 1.85 | 2.33 | 193.32 |
| VastGaussian$^{\dagger}$ | - | - | - | - | - | - | - |
| DoGaussian | 21.61 | 0.652 | 0.649 | 02:33 | 2.37 | 2.89 | 180.51 |

---

> ### Author Response · Authors · 2024-08-13
> **Minor Typo**
>
> There is a minor typo in our attached PDF: The bottom-left images should be swapped in Fig.2 of the attached file, and the full model handles the inconsistency issue in the division boundary. We will include the corrected results in the supplementary of our revised paper.
>
> Again, we thank all reviewers for their insightful suggestions and discussion.

---

### Decision · Program_Chairs · 2024-09-25

**Decision:**

Accept (poster)

**Comment:**

The paper received good scores, and the authors-reviewers engaged in a productive discussion during the rebuttal phase. The consensus is to accept the paper.

The paper presents an ADMM-based solution to split any large scene reconstruction into multiple small/local GS reconstruction tasks while maintaining consensus using a global GS solution using ADMM for synchronization. All the reviewers liked the approach and appreciated the 6x improvement in computation. Questions around the role of the global node and comparisons with alternatives (VastGS, etc.) were resolved in the rebuttal phase. Authors are encouraged to add additional clarifications while preparing the final version of the submission.